# Self-Assessment of Physical Fitness and Health versus Motivational Value of Physical Activity Goals in People Practicing Fitness, Football, Martial Arts and Wheelchair Rugby

**DOI:** 10.3390/ijerph191711004

**Published:** 2022-09-02

**Authors:** Katarzyna Kotarska, Celina Timoszyk-Tomczak, Leonard Nowak, Katarzyna Sygit, Izabela Gąska, Maria Alicja Nowak

**Affiliations:** 1Institute of Physical Culture Sciences, University of Szczecin, 71-065 Szczecin, Poland; 2Faculty of Social Sciences, Institute of Psychology, University of Szczecin, 71-017 Szczecin, Poland; 3Faculty of Health Sciences, Calisia University, 62-800 Kalisz, Poland; 4Faculty of Medical Science, Jan Grodek State University, 38-500 Sanok, Poland

**Keywords:** self-esteem, self-assessment, physical fitness, health, physical activity goals, motivation, IPAO

## Abstract

The aim of the study was to determine the relationship between self-assessment of physical fitness and health, and the motivational role of physical activity goals in people, depending on their sports discipline. The study included 470 men and 218 women, aged 18–45, from western and southern Poland. The respondents practiced sports recreationally (fitness—F), competitively (football—FB, martial arts—MA), and for rehabilitation and sports purposes (wheelchair rugby—R). The standardized questionnaire for the motivational role of physical activity goals (*Inventory of Physical Activity Objectives*, IPAO) by Lipowski and Zaleski and the authors’ questionnaire on lifestyle were used. In the statistical analyses, non-parametric statistics were used. Individuals with very high and high self-assessment of their physical fitness and very good self-assessment of health achieved higher scores on the motivational value scale, time management, motivational conflict and multidimensionality of physical activity goals (*p* < 0.05). Respondents who assessed their health as very good achieved lower results on the perseverance scale, compared to those who assessed their health as good or poor. Self-assessment of physical fitness had a positive, high and moderate correlation with the self-assessment of health in people practicing wheelchair rugby, fitness, football and martial arts (r = 0.61; r = 0.52; r = 0.41; r = 0.40, respectively). Correlations were found between the motivational role and time management in people practicing fitness (r = 0.81), football (r = 0.66) and martial arts (r = 0.45), and multidimensionality of goals in those practicing fitness (r = 0.65) and martial arts (r = 0.42) Wheelchair rugby players scored the highest on all KCAF scales (except for motivational conflict). Self-assessment of physical fitness of wheelchair rugby players and fitness practitioners was negatively correlated with time management (r = −0.68; r = −0.49), multidimensionality of goals (r = −0.51; r = −0.49) and motivational values (r = −0.43; r = −0.43). The demonstrated relationships indicated that there was a need to strengthen the self-esteem and motivation for physical activity, promoting perseverance, the ability to focus on the implementation of one’s goals and prioritizing the goals among people practicing various sports disciplines.

## 1. Introduction

Scientific research has repeatedly confirmed the relationship between physical activity and fitness, and health, as well as between physical activity and self-assessment of health and physical fitness [1,2,3,4,5,6,7,8]. According to M. Rosenberg, it is defined as a conscious, generalized (global) attitude towards one’s own self, which may be positive or negative [9,10,11]. Partial (specific) self-assessments of specific activities are diverse and can evolve, inter alia, under the influence of changes in health, physical fitness and life experiences. Individuals with high self-esteem accept themselves and do not mind making mistakes [10,12,13,14,15,16]. Compared to those with low self-esteem, they believe more strongly in their chances for success, are less discouraged by failures, and are more convinced of their own effectiveness [13]. According to the Exercise and Self-Esteem Model (EXSEM), the influence of physical activity on self-esteem is related to the hierarchical structure of self-esteem [17]. The experiences gained while making physical effort cause positive changes in the sense of competence in the physical sphere and increase the sense of one’s somatic acceptance. An increase in partial self-esteem related to physical activity, acceptance of one’s body and sense of fitness may translate into an increase in general self-esteem. Additionally, it increases the chances of getting involved in training, which results from increasingly higher competencies in the selected discipline. Supporting an individual in achieving their specific goals, encouraging them to increase their efforts and maintain their commitment to a given activity is one of the main functions of self-esteem [18]. Self-esteem and motivation influence human performance. They determine faith in one’s own abilities, help overcome obstacles and focus on achieving the determined goals [19]. Motivation gives energy to human behavior and sets a direction for it. Goals play a motivating role when they are of value to individuals, when they can be achieved through their own actions and when they are located in the long-term future [20,21]. The more chances of achieving goals, the greater their motivational force. Adequate self-esteem and appropriate motivation allow for achieving high sports results as one’s goals [22,23]. In order to achieve a specific goal, an individual focuses on it until he or she achieves it [3,24].

The purpose of the study (which used the Inventory of Physical Activity Objectives (IPAO), the Sense of Coherence Questionnaire, and the Connectedness to Nature Scale), was to explore the links between a sense of coherence and connectedness to nature in the context of motivation for 127 karate practitioners. Connectedness to nature correlated with motivational conflict positively in women and negatively in men. Understanding both the natural environment and the usefulness of setting sport-related goals led to increasing one’s efforts on the way to both successes and defeats, and, most of all, overcoming one’s weaknesses [25].

Bearing in mind the current knowledge about the motivation to undertake various forms of sports activity and self-assessment of health and physical fitness, the authors tried to determine how these relationships work in regard to people exercising recreationally and practicing competitive sports, as well as those combining rehabilitation with sports.

The aim of the study was to determine the relationship between self-assessment of physical fitness and health, and the motivational role of physical activity goals in people who practice sports recreationally and competitively and who combine sports with rehabilitation (fitness, football, martial arts, wheelchair rugby). 

It was hypothesized that there was a relationship between self-assessment of one’s fitness and health, and the motives for practicing sports. It was also assumed that this relationship could be modified by the type of sport practiced (fitness, football, martial arts and wheelchair rugby). 

## 2. Materials and Methods

### 2.1. Participants 

The study included 688 people with various levels of fitness and sports skills. The respondents came from sports and recreational clubs located in the western and southern parts of Poland. The respondents practiced sports recreationally (fitness), competitively (football, martial arts), and for rehabilitation and sports purposes (wheelchair rugby). The groups were diverse in terms of their sporting achievements. Among the women who practiced football, 14 played in the first league. Martial artists achieved over a dozen championship titles in the international (4) and national (10) arenas. Rugby players recently won the title of Polish champions. The majority of respondents were aged 20–29 (66.6%) and 68.3% of them were men. Almost 85% of the respondents lived in cities, with the exception of wheelchair rugby players (50% of whom lived in the countryside). Single persons accounted for 68.0%, while individuals in formal and informal relationships— 31%. The respondents had mainly above-secondary (56.5%) and secondary (38.3%) education. Over 50% of the respondents were professionally active, while over 40% were still studying. Only 8.0% of respondents were solely dependent on the disability benefit (most often wheelchair players).

### 2.2. Methods

A standardized questionnaire was used to analyze the motivational role of physical activity goals (*Inventory of Physical Activity Objectives*, IPAO) [26]. The term ‘physical activity’ included recreational activities as well as practicing competitive sports. The respondent determined the importance of individual goals on the Likert scale (1–5; 1 = “I do not agree at all” to 5 = “I fully agree”). Due to the underestimation of reliability by Cronbach alpha, McDonald’s ω was used, which was higher (Cronbach’s α = 0.78; McDonald’s ω = 0.85) in this study [27,28]. Based on factor analysis and matching of individual items in terms of the theory for the motivational role of goals, the following scales were distinguished: motivational values (McDonald’s ω = 0.83) (the force with which the goals influence the individual’s actions); time management (McDonald’s ω = 0.56) (the level of concentration on planning, organizing and subordinating time to physical activity); perseverance (McDonald’s ω = 0.64) (effectiveness and durability of action, and coping with obstacles); motivational conflict (McDonald’s ω = 0.70) (level of contradiction: PA goals vs other goals); multidimensionality of goals (McDonald’s ω = 0.81) (importance of the indicated goals). The authors’ questionnaire was also used, which included personal data, forms of recreational activity, types of sports disciplines, self-assessment of physical fitness and self-assessment of health. A five-point Likert scale was used in the self-assessment of physical fitness (very high, high, average, low, very low) and health (very good, good, average, poor, very poor).

In order to qualify people for the study, the authors used the following criteria: written informed consent to participate in the study; medically/legally significant degree of disability associated with diseases of the locomotor system—among wheelchair rugby players; regular participation in training (inclusion criteria); occasional participation in sports activities (less than once a week); participation in sports activities for less than six months; failure to specify the degree of disability; skipping answers (exclusion criteria). Before commencing the study, the consent of the trainers, instructors and club authorities was obtained, and the starting date was set for the study (preparatory period). The study was conducted by the authors of the presented article, as well as by trained instructors and trainers, using paper questionnaires.

### 2.3. Data Analysis

The study was based on a deliberate sample selection. Most variables displayed non-normal distribution; therefore, non-parametric statistics were used in the calculations. The analysis of the Kruskal–Wallis test (H test) and the U test were used to determine the significance of differences between the scales of motivational role, time management, perseverance, motivational conflict and multidimensionality goals in particular groups practicing individual sports. These methods were also used to analyze the relationship between self-assessment of physical fitness and health versus the motivational role of physical activity goals in people practicing fitness, football, martial arts and wheelchair rugby. The independence chi-square test was used to examine the relationship between two variables. The effect size was calculated for each test: E^2^R for the Kruskal–Wallis H test, Glass rank biserial correlation (rg) for the Mann–Whitney U test, Cramér’s V for the χ^2^ test. The value of *p* < 0.05 was assumed to be statistically significant. The relationships between fitness self-assessment and health self-assessment and the physical activity goals of recreational and competitive sports practitioners were calculated using intra-group correlations (r). Statistical calculations were made with Statistica 13.1 for Windows (StatSoft Sp. z o.o., Crakow, Poland), Microsoft Office Excel 2013 (Microsoft Sp. z o.o., Warsaw, Poland) and JASP 0.8.1.2 (https://jasp-stats.org (accessed on 25 April 2022)).

## 3. Results

The study involved 688 individuals practicing fitness (F), football (FB), martial arts (MA) and wheelchair rugby (R) (Table 1). Most of the respondents were men (68.3%). Wheelchair rugby was represented only by men. The average age of the respondents was 26.8 (minimum—18; maximum—48; SD—6.4). The respondents aged 20–29 accounted for 66.6%. Individuals aged 30 + (24%) included wheelchair rugby players (59.4%) and fitness practitioners (30.5%). Rural residents less frequently participated in training, with the exception of wheelchair rugby—half of the respondents practiced this sport (χ^2^ test = 36.03; *p* < 0.001; Cramérs V = 0.2). A total of 68% of respondents were single, while 31% remained in formal or informal relationships. The respondents had mainly secondary (38.3%) and above-secondary (56.5%) education. Wheelchair rugby players were more likely to have secondary education, but the differences were not significant. Some respondents still studied (26.5%), combined work and study (14.6%), worked intellectually (25.2%), physically (25.7%) or lived exclusively off disability benefits (8.0%). Wheelchairs rugby players more often worked intellectually and benefited from disability benefits, while fitness practitioners more often worked physically. More than half of the respondents assessed their financial situation as good (51%) and very good (31%). Football players (36.6%) and martial artists (30.9%) more often assessed their financial situation as very good, while those practicing rugby as sufficient (34.4%) (χ^2^ test = 84.73; *p* < 0.001; Cramérs V = 0.2).

The highest number of indications (on the Likert scale of 4 or 5 points) was achieved by the following goals: physical fitness, well-being, pleasure in physical activity, health and fulfilling the need for exercise. The goals that followed were related to the ability to relieve stress, maintain a slim figure, socialize, and escape from everyday life. The promotion of physical activity through one’s own example was of less importance. People practicing fitness, football and martial arts indicated the goals of physical activity in a similar order. For wheelchair rugby players, the following were more important: the pleasure of practicing sport, getting away from everyday life, the company of other people, and well-being (Table 2).

General differentiation was found in all motivational scales of the function of physical activity goals, depending on the sports discipline (H test, *p* < 0.05) (Table 3). 

Fitness practitioners achieved the lowest results in comparison with football players, martial artists and wheelchair rugby players on the following scales: motivational values time management multidimensionality of goals. Rugby players had better results compared to fitness practitioners, football players and martial artists on the time management scale, perseverance and multidimensionality of goals. The indicated differences had a strong and very strong effect, especially on the scale of multidimensionality of goals for rugby players (respectively: *p* < 0.001, *p* < 0.001, *p* < 0.001 for U-test; rg = −0.6, rg = −0.5, rg = −0.4). Martial artists achieved the highest results on the scale of motivational conflict in comparison with fitness practitioners and football players, but the differences showed a slight effect strength. Between football players and rugby players, there were differences in the time management scale, perseverance and multidimensionality of goals.

Descriptive statistics for self-assessment of physical fitness and health of the respondents are presented in Figure 1.

Fitness practitioners and rugby players had similar mean scores of self-assessments of their physical fitness, but rugby players had the lowest outliers and extremes, similarly to their health self-assessment scores. The average physical fitness and health self-assessments for soccer players were the lowest. Fitness practitioners had the widest range of values of self-assessment of physical fitness and health (there gave the most diverse assessments) (Figure 1). 

There was a general variation in the motivational function of physical activity goals (IPAO) depending on the self-assessment of physical fitness (H test, *p* < 0.05), with the exception of the perseverance scale (Table 4). Respondents who assessed their physical fitness as ‘very high’ and ‘high’ achieved higher results on individual scales compared to those who assessed their fitness as ‘average’ and ‘low’. Individuals who assessed their physical fitness as ‘very high’ compared to those who assessed their physical fitness as ‘high’, ‘average’ and ‘low’, achieved higher results on the following scales: motivational values, time management, motivational conflict, and multidimensionality of goals. Differences were also observed with regard to assessing physical fitness as ‘high’ compared to ‘average’ and ‘low’ physical fitness on the scales of motivational values and the multidimensionality of goals. On the time management scale, differences were found between people with ‘high’ and ‘average’ self-assessments, as well as in the multidimensionality of goals. Individuals with high self-assessments also had better results compared to those who assessed physical fitness as ‘average’ and ‘low’. On the motivational conflict scale, those with ‘low’ and ‘very low’ self-assessments of physical fitness differed from those with average self-assessments.

General differentiation was found on all scales related to physical activity goals, depending on the health self-assessment (H test, *p* < 0.05) (Table 5). The respondents who assessed their health as ‘very good’ achieved higher results compared to those assessing their health as ‘good’, ‘average’ or ‘poor’ on the scale of motivational values, time management and multidimensionality of goals. On the perseverance scale, respondents who assessed their health as ‘very good’ achieved lower results compared to those who assessed their health as ‘good’ or ‘poor’. On that scale, individuals who assessed their health as ‘poor’ had better results compared to those who assessed it as ‘good’ and ‘average’ (respectively: *p* < 0.070, *p* < 0.046 for the U test; rg = −0.4, rg = −0.4). Respondents who assessed their health as ‘very good’ achieved a higher level of contradiction in achieving the physical activity goals (motivational conflict) compared to those who assessed their health as ‘good’ and ‘average’. 

Among people practicing fitness, positive high correlations of motivational values, time management (r = 0.81) and multidimensionality of goals (r = 0.65) were observed. Time management also correlated with the multidimensionality of goals (r = 0.65) (Table 6). Positive and moderate correlations were found between the self-assessment of physical fitness and health (r = 0.52). Negative and moderate correlations concerned self-assessment of fitness and motivational values (r = −0.43), time management (r = −0.49) and multidimensionality of goals (r = −0.49) in people practicing fitness. Motivational values and time management of football players correlated positively and highly (r = 0.66), while they correlated moderately with motivational conflict (r = 0.49) and self-assessment of physical fitness and health (r = 0.41). In martial artists, positive moderate correlations of self-assessment of physical fitness and health (r = 0.40), motivational values and time organization (r = 0.45) as well as time management and multidimensionality of goals (r = 0.42) were observed. Between the self-assessment of physical fitness and health of wheelchair rugby players, there was a positive and high correlation (r = 0.61), a high negative correlation with time management (r = −0.68), a negative moderate correlation with the multidimensionality of goals (r = −0.51) and motivational values (−0.43), as well as a moderate positive correlation with motivational conflict (r = 0.44). A positive moderate correlation between motivational values and time management with the multidimensionality of goals (r = 0.40; r = 0.47) and perseverance with the motivational conflict (r = 0.44) was also found. 

## 4. Discussion

The authors’ own study partially confirmed the hypothesis about the existence of a relationship between the self-assessment of physical fitness, self-assessment of health and the motivational functions of sports goals. Respondents who assessed their physical fitness as ‘very high’ and ‘high’ achieved higher results in the motivational values scale, time management scale, motivational conflict scale and multidimensionality of goals scale, as compared to those who assessed their fitness as ‘average’ and ‘low’. In other studies, it was noticed that people with lower self-esteem were less motivated to undertake further activities, achieve goals and manage their time appropriately [29]. Research on judo players showed that practicing sports is an important factor contributing to the increase in the level of self-esteem; moreover, people with high self-esteem achieved better sports results [30]. It was empirically confirmed that people with high self-esteem more slowly withdrew from achieving the goal when it was threatened and put even more effort attempts to reach their goal [31]. The effectiveness of actions is fostered by the belief in self-efficacy, which is typical of people with high self-esteem [32]. The impact of physical activity on total self-esteem (improved physical fitness, physical competence and attractiveness), both in the short-term and long-term perspective, had been confirmed in a study on 235 adults and elderly Belgians [11]. High self-esteem was conducive to positive assessments of the physical condition of women who practiced aerobics, despite negative body self-assessments [33].

It was also confirmed that people who assessed their health as ‘very good’ on the scales of motivational values, time management and multidimensionality of goals, had better results compared to those who rated it lower. On the other hand, on the perseverance scale, individuals who assessed their health as ‘very good’ achieved lower results compared to those who assessed their health as ‘good’ or ‘poor’; those who assessed their health as ‘average’ also had lower scores on this scale compared to those who assessed it as ‘poor’. Lower results on the perseverance scale of people assessing their health as ‘very good’ and ‘average’ compared to those assessing it as ‘poor’ indicated lower effectiveness and perseverance, as well as greater difficulties in coping with adversities. Respondents who assessed their health as ‘very good’ also displayed a higher level of contradiction in the motivational conflict compared to those assessing their health as ‘good’ or ‘average’. This would mean that physical activity did not always win with other (non-sport) goals. People in very good health who practice sports are less persistent and determined to overcome various adversities. Presumably, they have more available alternatives, and a lower self-assessment of their own health is a limitation—a challenge that promotes greater perseverance. This may be explained by the mechanism of reactance when we try to regain the freedom of choice when we feel limited and therefore we are more determined to act [34]. In studies on the relationship between physical activity, motivation and mental well-being of young adults, it was found that a higher level of physical activity is associated with better mental well-being. An important determinant of mental well-being was intrinsic motivation [35]. People with high expectations of self-efficacy are healthier and more successful [36,37]. Not all studies show a relationship between recreational/sports activity and self-esteem. In a study by Dubielis (2021) on 243 students of senior secondary schools, half of whom attended only Physical Education lessons, no relationship between self-esteem and physical activity was found. Studies on 250 students of physiotherapy at the Medical University of Silesia (SUM) in Katowice also did not confirm this relationship. The respondents were characterized by low self-esteem. The high level of physical activity of students (related to the field of study) did not increase their self-esteem. This result may be due to the underestimation of the value of physical activity and a healthy lifestyle [38]. In light of the Ho T-W (2020) study, insufficient awareness of the importance of physical activity and fitness for health may cause difficulties in self-assessment [39].

This study found that rugby players had the best results in IPAO on all scales (motivational values, time management, perseverance and multidimensionality of goals), with the exception of motivational conflict (characterized by a similar level of contradiction between the goals of physical activity and non-sport goals), in which they did not differ from the other groups. These differences may result from the fact that rugby players focused on their training and organized their lives around them. Compared to the motivational functions of the goals of karate practitioners, boxers, football players and wheelchair basketball players (amateur level), no differences were observed on the perseverance scale, which was correlated only with sports experience [39,40,41]. 

Among examined women, no differences were observed in the scales determining the motivational functions of the aims (IPAO) of participation in combat sports, martial arts and fitness training. However, an overall differentiation in terms of motivational values and time management depending on the training experience of respondents was reported [42].

The motivation of people with disabilities who practiced sports did not differ from that of non-disabled athletes; the surveyed rugby players (who achieved high results at the national level) displayed higher motivation. Sports activity of people with disabilities is also important in experiencing life satisfaction, personal development, and changes in the way they perceive themselves [41,42,43]. 

This study found a positive strong correlation between self-assessment of the physical fitness and health of wheelchair rugby players, as well as moderate correlations among fitness, football and martial arts practitioners. Self-assessment of physical fitness of wheelchair rugby players and fitness practitioners was negatively (highly and moderately) correlated with time management, the multidimensionality of goals and motivational values. The authors also observed a positive, moderate correlation between perseverance and motivational conflict among rugby players and fitness practitioners. These results show that with the increase in fitness self-assessment, rugby players and fitness practitioners had fewer difficulties in staying motivated, focusing on achieving goals related to physical activity, being more persistent, and managing their time better for recreational/sports activity. In a study of 375 English people practicing recreational and health-promoting exercises in clubs, it was found that individuals undertaking physical activity showed much greater motivation to exercise than people at the preparation and action stages. Positive, high correlations were also found between the motivational values and the time management of people practicing fitness and football, as well as average correlations for martial artists. On the time management scale, there were positive high correlations with the multidimensionality of goals for fitness practitioners, and positive moderate correlations for martial artists and wheelchair rugby players. On the motivational conflict scale, moderate positive correlations between the motivational values of fitness protocones and football players were found [44]. The research confirmed the role of motivation for participating in exercises depending on, e.g., the type of activity. The importance of martial arts in acquiring life skills, especially time management, communication and leadership factors, was demonstrated in the example of judo players (n = 947) [45,46]. 

The relationships between self-assessment of health and physical fitness as well as motivational functions of recreational/sports activity goals indicate that there are still opportunities and need to strengthen self-esteem and motivation for physical activity, also among those who are physically active (e.g., through the active role of trainers, sports instructors, sports psychologists, health education). In shaping the motivation to practice sport, one should engage in helping players remove obstacles that make it difficult to achieve the goal, focusing attention on the goal and building the conviction that this goal is possible to achieve [10,40]. People with disabilities require special assistance, especially physically inactive people who avoid rehabilitation [47,48,49].

## 5. Conclusions

The demonstrated correlations of self-assessment of physical fitness and health and motivational functions of goals in people practicing fitness, football, martial arts and wheelchair rugby, indicate the need to strengthen the self-esteem and motivation for physical activity, shape perseverance, the ability to focus on the implementation of goals and their hierarchization, incl. the active role of coaches, sports instructors, physiotherapists, sports psychologists, and health educators.

## 6. Limitations

The demonstrated relationships between self-assessment of physical fitness and health with the motivational functions of physical activity goals do not fully exhaust the subject matter of this study, given the cross-sectional nature of the study and the inability to establish cause-and-effect relationships. In future studies, it seems advisable to relate self-assessment and motivation to years of experience in physical activity and respondents’ awareness of the importance of physical activity in shaping physical fitness and health. It is also necessary to include women in all surveyed sports disciplines, especially those related to the sport of people with disabilities. Taking these relationships into account shall allow for more detailed identification of the determinants of self-esteem and motivational functions of the physical activity goals.

## Figures and Tables

**Figure 1 ijerph-19-11004-f001:**
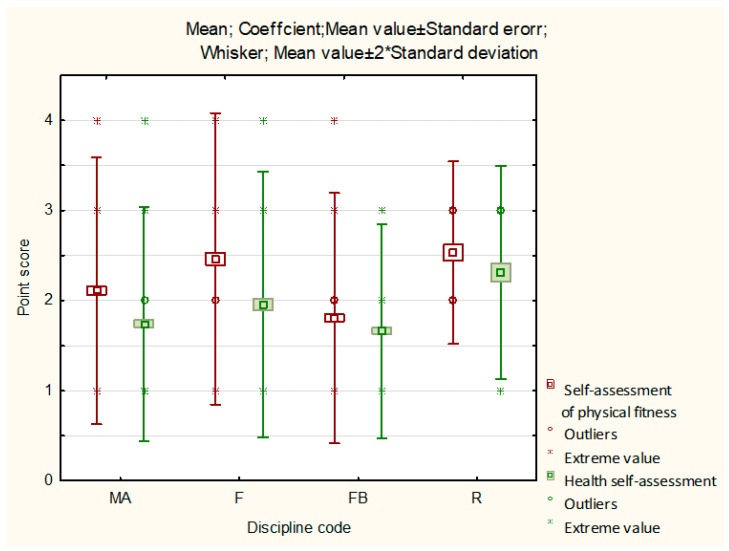
Descriptive statistics for self-assessment of physical fitness and health depending on the sports discipline.

**Table 1 ijerph-19-11004-t001:** Socio-economic characteristics of people practicing sports (n = 688).

Variables	Sports Discipline				Total (N = 688)	
	Fitness (n = 141)	Football (n = 278)	Martial Arts (n = 237)	Wheelchair Rugby (n = 32)	N	%
Gender:						
-male	66.0	77.7	54.4	100.0	470	68.3
-female	34.0	22.3	45.6	-	218	31.7
Age:						
<20 years	6.4	10.4	11.4	-	65	9.4
20–24	22.7	43.5	32.1	-	229	33.3
25–29	23.4	38.5	34.6	21.9	229	33.3
30–34	17.0	6.5	14.3	18.7	82	11.9
≥35	30.5	1.1	7.6	59.4	83	12.1
Place of residence:						
-urban	83.0	83.1	90.7	50.0	579	84.2
-rural	17.0	16.9	9.3	50.0	109	15.8
Marital status: (n = 678)						
-I’m not in a relationship	53.3	77.7	67.5	50.0	461	68.0
-I’m married	42.2	17.2	22.8	37.5	170	25.1
-I’m divorced	0.8	-	1.3	9.4	7	1.0
-I have a partner (cohabitation)	3.7	5.1	8.4	3.1	40	5.9
Education:						
-below secondary	4.3	4.0	6.8	6.7	35	5.2
-secondary	31.4	39.3	38.3	60.0	259	38.3
-above secondary	64.3	56.6	54.9	33.3	383	56.5
Types of activity:						
-studying	18.6	36.1	22.4	9.4	182	26.5
-physical work	38.6	25.6	21.5	-	176	25.7
-intellectual work	27.1	17.3	28.7	59.4	173	25.2
-study and work	13.6	11.6	19.4	9.4	100	14.6
-pension	2.1	9.4	8.0	21.8	55	8.0
Financial situation:						
-very good	26.4	36.6	30.9	3.1	210	31.0
-good	51.4	52.8	47.2	62.5	346	51.0
-sufficient	22.2	10.6	21.9	34.4	122	18.0

**Table 2 ijerph-19-11004-t002:** The most important goals for physical activity (IPAO) (%).

Goals of Physical Activity		Sports Discipline				Total (688)	
		Fitness	Football	Martial Arts	Wheelchair Rugby	N	%
1	Health	44.3	39.6	48.1	18.8	291	42.5
2	Physical fitness, good condition	20.4	40.5	34.5	4.7	407	59.2
3	Company of other people	22.5	30.9	35.0	71.9	223	32.6
4	Slim body	32.4	34.9	38.0	6.3	234	34.1
5	Well-being	52.5	48.9	63.7	71.9	383	55.8
6	Being physically active and fit	12.9	14.0	21.2	-	107	15.6
7	Higher self-esteem	13.6	18.7	15.6	34.4	119	17.3
8	Pleasure	44.3	47.5	58.5	78.1	357	52.0
9	Escape from everyday life	26.4	21.6	41.3	71.9	217	31.7
10	Stress relief	34.5	28.1	48.7	50.0	257	37.5
11	Satisfying the need for exercise	37.1	37.0	4.3	62.5	287	41.8
12	Promoting physical activity	18.6	24.1	34.6	-	175	25.5

**Table 3 ijerph-19-11004-t003:** Motivational role of physical activity goals (IPAO), depending on the sport discipline (H test, E^2^R, U test, rg).

Specification	Sports Discipline	FB	MA	R	FB	MA	R	Rank Means
		*p* Value for U Statistics			Glass Rank Biserial Correlation (rg)			
Motivational value	F	0.001	0.000	0.000	−0.2	−0.3	−0.4	279.4
H (3, N = 688) = 21.37	FB		0.401	0.265		−0.1	−0.1	351.3
E^2^_R_ = 0.03	MA			0.257			−0.1	367.2
*p* < 0.000	R							402.9
Time management	F	0.006	0.000	0.000	−0.2	−0.2	−0.4	290.7
H (3, N = 688) = 18.90	FB		0.392	0.015		−0.1	−0.3	346.7
E^2^_R_ = 0.03	MA			0.039			−0.2	361.6
*p* < 0.000	R							435.3
Perseverance	F	0.750	0.710	0.015	0.1	0.1	−0.3	345.4
H (3, N = 688) = 10.33	FB		0.893	0.001		0.1	−0.4	338.3
E^2^_R_ = 0.01	MA			0.001			−0.3	336.5
*p* < 0.015	R							453.1
Motivational conflict	F	0.363	0.003	0.256	−0.1	−0.2	−0.1	314.1
H (3, N = 688) = 10.24	FB		0.015	0.549		−0.1	−0.1	332.7
E^2^_R_ = 0.01	MA			0.454			0.1	375.2
*p* < 0.016	R							352.8
Multidimensionality of goals	F	0.000	0.000	0.000	−0.2	−0.3	−0.6	276.3
H (3, N = 688) = 38.48	FB		0.144	0.000		−0.1	−0.5	342.3
E^2^_R_ = 0.06	MA			0.000			−0.4	366.9
*p* < 0.000	R							496.8

F—fitness; FB—football; MA—martial arts; R—wheelchair.

**Table 4 ijerph-19-11004-t004:** Motivational functions of physical activity goals (IPAO), depending on the self-assessment of physical fitness of the respondents (H test, E 2 R, U test, rg).

Specification	Physical Fitness	High	Average	Low	High	Average	Low	Rank Means
		*p* Value for U Statistics			Glass Rank Biserial Correlation (rg)			
Motivational value	Very high	0.000	0.000	0.000	0.3	0.5	0.5	444.4
H (3, N = 687) = 79.86	High		0.000	0.086		0.3	0.3	346.2
E^2^_R_ = 0.11	Average			0.916			0.1	259.4
*p* < 0.000	Low and very low							256.2
Time management	Very high	0.000	0.000	0.013	0.4	0.6	0.4	458.9
H (3, N = 687) = 107.34	High		0.000	0.628		0.3	0.1	346.8
E^2^_R_ = 0.15	Average			0.188			−0.2	240.8
*p* < 0.000	Low and very low							319.9
Motivational conflict	Very high	0.027	0.008	0.006	0.1	0.2	0.4	381.5
H (3, N = 687) = 13.47	High		0.388	0.016		0.1	0.4	341.0
E^2^_R_ = 0.02	Average			0.046			0.3	325.8
*p* < 0.003	Low and very low							223.2
Multidimensionality of goals	Very high	0.000	0.000	0.000	0.3	0.5	0.7	439.2
H (3, N = 687) = 71.01	High		0.000	0.002		0.2	0.5	343.9
E^2^_R_ = 0.10	Average			0.114			0.3	272.9
*p* < 0.000	Low and very low							193.6

**Table 5 ijerph-19-11004-t005:** Motivational functions of physical activity goals (IPAO), depending on the self-assessment of health of the respondents (H test, E 2 R, U test, rg).

Specification	Health Self-Assessment	Good	Average	Poor	Good	Average	Poor	Rank Means
		*p* Value for U Statistics			Glass Rank Biserial Correlation (rg)			
Motivational value	Very good	0.000	0.000	0.018	0.3	0.4	0.5	413.4
H (3, N = 687) = 50.58	Good		0.008	0.333		0.2	0.2	318.7
E^2^_R_ = 0.07	Average			0.925			−0.1	254.6
*p* < 0.001	Poor and very poor							249.8
Time management	Very good	0.000	0.000	0.012	0.3	0.4	0.5	417.2
H (3, N = 687) = 52.91	Good		0.087	0.324		0.1	0.2	314.2
E^2^_R_ = 0.07	Average			0.838			0.1	267.5
*p* < 0.001	Poor and very poor							243.2
Perseverance	Very good	0.045	0.143	0.042	−0.1	−0.1	−0.4	320.4
H (3, N = 687) = 8.20	Good		0.960	0.070		0.0	−0.4	353.6
E^2^_R_ = 0.01	Average			0.046			−0.4	356.6
*p* < 0.041	Poor and very poor							477.8
Motivational conflict	Very good	0.001	0.003	0.158	0.2	0.2	0.3	382.0
H (3, N = 687) = 14.81	Good		0.338	0.477		0.1	0.1	328.9
E^2^_R_ = 0.02	Average			0.750			0.1	303.6
*p* < 0.002	Poor and very poor							280.2
Multidimensionality of goals	Very good	0.000	0.000	0.001	0.3	0.6	0.7	418.2
H (3, N = 687) = 66.18	Good		0.000	0.096		0.3	0.3	322.2
E^2^_R_ = 0.09	Average			0.905			−0.1	224.3
*p* < 0.001	Poor and very poor							204.3

**Table 6 ijerph-19-11004-t006:** Intra-group correlation coefficients (r) of self-assessment of physical fitness, health and motivational goals (IPAO) in people practicing fitness, football, martial arts and wheelchair rugby.

Sports Discipline	Specification	Intra-Group Correlations					
		2	3	4	5	6	7
Fitness	1. Self-assessment of physical fitness	0.52 *	−0.43 *	−0.49 *	0.05	−0.19 *	−0.49 *
	2. Self-assessment of health	-	−0.31 *	−0.34 *	0.08	−0.14	−0.38 *
	3. Motivational values	-	-	0.81 *	0.05	0.45 *	0.65 *
	4. Time management	-	-	-	0.13	0.42 *	0.65 *
	5. Perseverance	-	-	-	-	0.41 *	0.20 *
	6. Motivational conflict	-	-	-	-	-	0.33 *
	7. Multidimensionality of goals	-	-	-	-	-	-
Football	1. Self-assessment of physical fitness	0.41 *	−0.29 *	−0.23 *	0.05	−0.11	−0.32 *
	2. Self-assessment of health	-	−0.26 *	−0.21 *	0.05	−0.10	−0.38 *
	3. Motivational values	-	-	0.66 *	0.02	0.49 *	0.32 *
	4. Time management	-	-	-	0.13 *	0.36 *	0.31 *
	5. Perseverance	-	-	-	-	0.26 *	0.03
	6. Motivational conflict	-	-	-	-	-	0.19 *
	7. Multidimensionality of goals	-	-	-	-	-	-
Martial arts	1. Self-assessment of physical fitness	0.40 *	−0.14 *	−0.38 *	−0.06	−0.11	−0.24 *
	2. Self-assessment of health	-	−0.13 *	−0.18 *	0.03	−0.21 *	−0.27 *
	3. Motivational values	-	-	0.45 *	−0.23 *	0.02	0.31 *
	4. Time management	-	-	-	0.15 *	0.03	0.42 *
	5. Perseverance	-	-	-	-	0.24 *	0.20 *
	6. Motivational conflict	-	-	-	-	-	0.12
	7. Multidimensionality of goals	-	-	-	-	-	-
Wheelchair rugby	1. Self-assessment of physical fitness	0.61 *	−0.43 *	−0.68 *	0.34 *	0.44 *	−0.51 *
	2. Self-assessment of health	-	−0.35 *	−0.38 *	0.28	0.29	−0.37 *
	3. Motivational values	-	-	0.33	−0.32	−0.17	0.40 *
	4. Time management	-	-	-	0.10	−0.10	0.47 *
	5. Perseverance	-	-	-	-	0.44	−0.20 *
	6. Motivational conflict	-	-	-	-	-	−0.15 *
	7. Multidimensionality of goals	-	-	-	-	-	-

* statistically significant for *p* < 0.05.

## Data Availability

Data are not publicly available and data sharing is not applicable to this article.

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
