# Peer review of "Self-Assessment of Physical Fitness and Health versus Motivational Value of Physical Activity Goals in People Practicing Fitness, Football, Martial Arts and Wheelchair Rugby"

_ijerph, 2022, doi:10.3390/ijerph191711004_

Round 1

Reviewer 1 Report

It was a pleasure to review the manuscript.

This paper by Katorska and coworkers deals with an interesting topic on self-assessment of physical fitness and health versus the motivation value of physical activity goals. The paper posed to determine the difference between people playing different kinds of sports disciplines.

The abstract is too long (around 400 words) and gets lost in the details.

The introduction section provides sufficient background on EXEM but the terminology used is not always accurate. Although the questionnaire measures physical activity, the manuscript rather examines the role of sports, which is a confusion of concepts. More precise use of terms and definitions is needed.

There is redundancy in the materials and methods and results section regarding the description of the population, which should be eliminated.

The data collection procedure is not enough detailed: details on who, how, where and when invited the participants, whether they fill the questionnaires in paper-pencil or online form with or without help etc.

The study design is also unclear. Please make more methodological sense of study design, e.g. cross-sectional study with convenient sampling.

The ethical issues related to the back matters, it should be deleted from the materials and methods section.  

Titles of the tables are not clear without the context of the manuscript, please add the necessary information e.g. the IPAO was used as the measurement tool. Please add the meaning of the abbreviations e.g. F, FB, MA, R... in the footnotes of the tables.

In Table 1. the first age range is not proper, <20 would be more precise instead of ≤20. Pupil work sounds strange, please revise. Pensioners (not working) are mentioned in the table but disability benefit is in the text, please unify. Please revise the title, there are no data on the independence χ2 test, and Cramér's V, is only mentioned in the text.

Is Figure 1. automatically generated by one of the statistical software? Please add a more sophisticated one.

In general, the result section is not easy to follow, and the content is conflicting with the conclusion of the manuscript, which is my major concern. The statement, that „Authors’ own study partially confirmed the hypothesis about the existence of a relationship between the self-assessment of physical fitness, self-assessment of health and the motivational functions of sports goals.” should be considered excessive, because of the negative correlation found between the motivational values and self-assessment of health or motivation and self-assessment of physical fitness. (Table 6.)

The importance of the results could b better assessed if the significance levels were given in addition to the R values for the correlation

In the discussion section, we can not read about studies using the same tool, the IPAO, which would give a good comparison. Including additional references would be useful, because of the low ratio of recent citations (only 27% within 5 years).

The limitation section is modest and does not mention e.g. the method of sampling, proportion of genders, comparison of people with and without disability, and self-report on health status and physical fitness, which can lead to possible bias.

Wheelchair basketball and rowing are mentioned in the acknowledgment section, which are not included in the study.

The manuscript is suitable for publication after major revision.

Reviewer 2 Report

Dear Editor,

Thank you for allowing me to review the article entitled "Self-assessment of physical fitness and health versus motivational value of physical activity goals in people practicing fitness, football, martial arts and wheelchair rugby". The study analyzes the relationship between self-assessment of physical activity and health and motivation to practice physical activity in athletes of different disciplines and category. The manuscript makes an adequate review of previous studies, theoretical foundation and exposition of methodology and results obtained. The conclusions may be of interest to the readers of the journal and for this reason I recommend its publication after the modification of certain minors concerns:

1. In the Introduction, the relationship between self-esteem of physical activity and general self-esteem is explained, as well as the relationship between motivation and performance of physical activity. But I suggest the authors to expand, theoretically and based on previous studies, the possible relationship between self-esteem and motivation in the practice of physical activities to support their hypotheses.

2. In the Method, the qualitative values of the Likert scales used in each of the questionnaires should be given in order to better understand the construct evaluated. In addition, the reliability values of all the scales used, previously validated or created for this study, should be detailed. The results of the factor analysis and the theoretical bases of the subdimensions in previous studies should also be specified.

3. In the Results section: 1) Give data on minimum and maximum age as well as mean and standard deviation. 2) The same statistical information is given in the tables as in the text, eliminate one or the other, I suggest keeping the tables and point out only where differences are found and the sense in the text. 3) In addition, it is not understood why in tables 2-4 the comparisons by groups are given only in terms of p value, because in this way it is not known where there are significant differences who scores more or less on the scale: I suggest that in the tables of comparison by group of sporting activity should give the total summed scores of each group on the scale or the mean, always including the standard deviation. 4) In table 6 indicate significant results with asterisks the significance of  p-values. 5) In all tables put a legend at the bottom with the full names of the acronyms used as well as detailing when percentages or counted scores are used.

4. In the Discussion: 1) It is assumed in the manuscript that a higher self-esteem for physical activity and health influences the motivation to practice sports; however, this is not justified because this relationship could not be in the opposite direction: the higher the motivation to practice physical activity, the higher the self-esteem? 2) It is stated that "This study found that rugby players had the best results in IPAO on all scales (motivational values, time management, perseverance and multidimensionality of goals), ...", however, the rugby players were all men, so there may be a confusion between the variables when explaining the results. It would be necessary to provide data on the difference by sex in each scale and see if these may not be accounting for part of the variance explained. It would also be necessary to obtain the correlation of age with each of the measures to rule out the effect of age.
